# Learning word embeddings efficiently with noise-contrastive estimation

**Andriy Mnih**
DeepMind Technologies
andriy@deepmind.com

**Koray Kavukcuoglu**
DeepMind Technologies
koray@deepmind.com

## Abstract

Continuous-valued word embeddings learned by neural language models have recently been shown to capture semantic and syntactic information about words very well, setting performance records on several word similarity tasks. The best results are obtained by learning high-dimensional embeddings from very large quantities of data, which makes scalability of the training method a critical factor.

We propose a simple and scalable new approach to learning word embeddings based on training log-bilinear models with noise-contrastive estimation. Our approach is simpler, faster, and produces better results than the current state-of-the-art method. We achieve results comparable to the best ones reported, which were obtained on a cluster, using four times less data and more than an order of magnitude less computing time. We also investigate several model types and find that the embeddings learned by the simpler models perform at least as well as those learned by the more complex ones.

## 1 Introduction

Natural language processing and information retrieval systems can often benefit from incorporating accurate word similarity information. Learning word representations from large collections of unstructured text is an effective way of capturing such information. The classic approach to this task is to use the word space model, representing each word with a vector of co-occurrence counts with other words [16]. Representations of this type suffer from data sparsity problems due to the extreme dimensionality of the word count vectors. To address this, Latent Semantic Analysis performs dimensionality reduction on such vectors, producing lower-dimensional real-valued word embeddings.

Better real-valued representations, however, are learned by neural language models which are trained to predict the next word in the sentence given the preceding words. Such representations have been used to achieve excellent performance on classic NLP tasks [4, 18, 17]. Unfortunately, few neural language models scale well to large datasets and vocabularies due to use of hidden layers and the cost of computing normalized probabilities.

Recently, a scalable method for learning word embeddings using light-weight tree-structured neural language models was proposed in [10]. Although tree-structured models can be trained quickly, they are considerably more complex than the traditional (flat) models and their performance is sensitive to the choice of the tree over words [13]. Inspired by the excellent results of [10], we investigate a simpler approach based on noise-contrastive estimation (NCE) [6], which enables fast training without the complexity of working with tree-structured models. We compound the speedup obtained by using NCE to eliminate the normalization costs during training, by using very simple variants of the log-bilinear model [14], resulting in parameter update complexity linear in the word embedding dimensionality.

We evaluate our approach on two analogy-based word similarity tasks [11, 10] and show that despite the considerably shorter training times our models outperform the Skip-gram model from [10] trained on the same 1.5B-word Wikipedia dataset. Furthermore, we can obtain performance comparable to that of the huge Skip-gram and CBOW models trained on a 125-CPU-core cluster after training for only four days on a single core using four times less training data. Finally, we explore several model architectures and discover that the simplest architectures learn embeddings that are at least as good as those learned by the more complex ones.

## 2 Neural probabilistic language models

Neural probabilistic language models (NPLMs) specify the distribution for the target word $w$, given a sequence of words $h$, called the context. In statistical language modelling, $w$ is typically the next word in the sentence, while the context $h$ is the sequence of words that precede $w$. Though some models such as recurrent neural language models [9] can handle arbitrarily long contexts, in this paper, we will restrict our attention to fixed-length contexts. Since we are interested in learning word representations as opposed to assigning probabilities to sentences, we do not need to restrict our models to predicting the next word, and can, for example, predict $w$ from the words surrounding it as was done in [4].

Given a context $h$, an NPLM defines the distribution for the word to be predicted using the scoring function $s_\theta(w, h)$ that quantifies the compatibility between the context and the candidate target word. Here $\theta$ are model parameters, which include the word embeddings. The scores are converted to probabilities by exponentiating and normalizing:

$$P_\theta^h(w) = \frac{\exp(s_\theta(w, h))}{\sum_{w'} \exp(s_\theta(w', h))}.$$ (1)

Unfortunately both evaluating $P_\theta^h(w)$ and computing the corresponding likelihood gradient requires normalizing over the entire vocabulary, which means that maximum likelihood training of such models takes time linear in the vocabulary size, and thus is prohibitively expensive for all but the smallest vocabularies.

There are two main approaches to scaling up NPLMs to large vocabularies. The first one involves using a tree-structured vocabulary with words at the leaves, resulting in training time logarithmic in the vocabulary size [15]. Unfortunately, this approach is considerably more involved than ML training and finding well-performing trees is non-trivial [13]. The alternative is to keep the model but use a different training strategy. Using importance sampling to approximate the likelihood gradient was the first such method to be proposed [2, 3], and though it could produce substantial speedups, it suffered from stability problems. Recently, a method for training unnormalized probabilistic models, called noise-contrastive estimation (NCE) [6], has been shown to be a stable and efficient way of training NPLMs [14]. As it is also considerably simpler than the tree-based prediction approach, we use NCE for training models in this paper. We will describe NCE in detail in Section 3.1.

## 3 Scalable log-bilinear models

We are interested in highly scalable models that can be trained on billion-word datasets with vocabularies of hundreds of thousands of words within a few days on a single core, which rules out most traditional neural language models such as those from [1] and [4]. We will use the log-bilinear language model (LBL) [12] as our starting point, which unlike traditional NPLMs, does not have a hidden layer and works by performing linear prediction in the word feature vector space. In particular, we will use a more scalable version of LBL [14] that uses vectors instead of matrices for its context weights to avoid the high cost of matrix-vector multiplication. This model, like all other models we will describe, has two sets of word representations: one for the target words (i.e. the words being predicted) and one for the context words. We denote the target and the context representations for word $w$ with $q_w$ and $r_w$ respectively. Given a sequence of context words $h = w_1, .., w_n$, the model computes the predicted representation for the target word by taking a linear combination of the context word feature vectors:

$$\hat{q}(h) = \sum_{i=1}^{n} c_i \odot r_{w_i},$$ (2)

where $c_i$ is the weight vector for the context word in position $i$ and $\odot$ denotes element-wise multiplication. The context can consist of words preceding, following, or surrounding the word being predicted. The scoring function then computes the similarity between the predicted feature vector and one for word $w$:

$$s_\theta(w, h) = \hat{q}(h)^\top q_w + b_w, \tag{3}$$

where $b_w$ is a bias that captures the context-independent frequency of word $w$. We will refer to this model as **vLBL**, for vector LBL.

vLBL can be made even simpler by eliminating the position-dependent weights and computing the predicted feature vector simply by averaging the context word feature vectors: $\hat{q}(h) = \frac{1}{n} \sum_{i=1}^{n} r_{w_i}$. The result is something like a local topic model, which ignores the order of context words, potentially forcing it to capture more semantic information, perhaps at the expense of syntax. The idea of simply averaging context word feature vectors was introduced in [8], where it was used to condition on large contexts such as entire documents. The resulting model can be seen as a non-hierarchical version of the CBOW model of [10].

As our primary concern is learning word representations as opposed to creating useful language models, we are free to move away from the paradigm of predicting the target word from its context and, for example, do the reverse. This approach is motivated by the distributional hypothesis, which states that words with similar meanings often occur in the same contexts [7] and thus suggests looking for word representations that capture their context distributions. The inverse language modelling approach of learning to predict the context from the word is a natural way to do that. Some classic word-space models such as HAL and COALS [16] follow this approach by representing the context distribution using a bag-of-words but they do not learn embeddings from this information.

Unfortunately, predicting an $n$-word context requires modelling the joint distribution of $n$ words, which is considerably harder than modelling the distribution of a single word. We make the task tractable by assuming that the words in different context positions are conditionally independent given the current word $w$:

$$P_\theta^w(h) = \prod_{i=1}^{n} P_{i,\theta}^w(w_i). \tag{4}$$

Though this assumption can be easily relaxed without giving up tractability by introducing some Markov structure into the context distribution, we leave investigating this direction as future work. The context word distributions $P_{i,\theta}^w(w_i)$ are simply vLBL models that condition on the current word and are defined by the scoring function

$$s_{i,\theta}(w_i, w) = (c_i \odot r_w)^\top q_{w_i} + b_{w_i}. \tag{5}$$

The resulting model can be seen as a Naive Bayes classifier parameterized in terms of word embeddings. As this model performs inverse language modelling, we will refer to it as **ivLBL**.

As with our traditional language model, we also consider the simpler version of this model without position-dependent weights, defined by the scoring function

$$s_{i,\theta}(w_i, w) = r_w^\top q_{w_i} + b_{w_i}. \tag{6}$$

The resulting model is the non-hierarchical counterpart of the Skip-gram model [10]. Note that unlike the tree-based models, such as those in the above paper, which only learn conditional embeddings for words, in our models each word has both a conditional and a target embedding which can potentially capture complementary information. Tree-based models replace target embeddings with parameters vectors associated with the tree nodes, as opposed to individual words.

## 3.1 Noise-contrastive estimation

We train our models using noise-contrastive estimation, a method for fitting unnormalized models [6], adapted to neural language modelling in [14]. NCE is based on the reduction of density estimation to probabilistic binary classification. The basic idea is to train a logistic regression classifier to discriminate between samples from the data distribution and samples from some "noise" distribution, based on the ratio of probabilities of the sample under the model and the noise distribution. The

main advantage of NCE is that it allows us to fit models that are not explicitly normalized making the training time effectively independent of the vocabulary size. Thus, we will be able to drop the normalizing factor from Eq. 1, and simply use $\exp(s_\theta(w, h))$ in place of $P_\theta^h(w)$ during training. The perplexity of NPLMs trained using this approach has been shown to be on par with those trained with maximum likelihood learning, but at a fraction of the computational cost.

Suppose we would like to learn the distribution of words for some specific context $h$, denoted by $P^h(w)$. To do that, we create an auxiliary binary classification problem, treating the training data as positive examples and samples from a noise distribution $P_n(w)$ as negative examples. We are free to choose any noise distribution that is easy to sample from and compute probabilities under, and that does not assign zero probability to any word. We will use the (global) unigram distribution of the training data as the noise distribution, a choice that is known to work well for training language models. If we assume that noise samples are $k$ times more frequent than data samples, the probability that the given sample came from the data is $P^h(D = 1|w) = \frac{P_d^h(w)}{P_d^h(w) + kP_n(w)}$. Our estimate of this probability is obtained by using our model distribution in place $P_d^h$:

$$P^h(D = 1|w, \theta) = \frac{P_\theta^h(w)}{P_\theta^h(w) + kP_n(w)} = \sigma\left(\Delta s_\theta(w, h)\right), \tag{7}$$

where $\sigma(x)$ is the logistic function and $\Delta s_\theta(w, h) = s_\theta(w, h) - \log(kP_n(w))$ is the difference in the scores of word $w$ under the model and the (scaled) noise distribution. The scaling factor $k$ in front of $P_n(w)$ accounts for the fact that noise samples are $k$ times more frequent than data samples.

Note that in the above equation we used $s_\theta(w, h)$ in place of $\log P_\theta^h(w)$, ignoring the normalization term, because we are working with an unnormalized model. We can do this because the NCE objective encourages the model to be approximately normalized and recovers a perfectly normalized model if the model class contains the data distribution [6].

We fit the model by maximizing the log-posterior probability of the correct labels $D$ averaged over the data and noise samples:

$$\begin{aligned} J^h(\theta) &= E_{P_d^h}\left[\log P^h(D = 1|w, \theta)\right] + kE_{P_n}\left[\log P^h(D = 0|w, \theta)\right] \\ &= E_{P_d^h}\left[\log \sigma\left(\Delta s_\theta(w, h)\right)\right] + kE_{P_n}\left[\log\left(1 - \sigma\left(\Delta s_\theta(w, h)\right)\right)\right], \end{aligned} \tag{8}$$

In practice, the expectation over the noise distribution is approximated by sampling. Thus, we estimate the contribution of a word / context pair $w, h$ to the gradient of Eq. 8 by generating $k$ noise samples $\{x_i\}$ and computing

$$\frac{\partial}{\partial\theta}J^{h,w}(\theta) = (1 - \sigma\left(\Delta s_\theta(w, h)\right))\frac{\partial}{\partial\theta}\log P_\theta^h(w) - \sum_{i=1}^{k}\left[\sigma\left(\Delta s_\theta(x_i, h)\right)\frac{\partial}{\partial\theta}\log P_\theta^h(x_i)\right]. \tag{9}$$

Note that the gradient in Eq. 9 involves a sum over $k$ noise samples instead of a sum over the entire vocabulary, making the NCE training time linear in the number of noise samples and independent of the vocabulary size. As we increase the number of noise samples $k$, this estimate approaches the likelihood gradient of the normalized model, allowing us to trade off computation cost against estimation accuracy [6].

NCE shares some similarities with a training method for non-probabilistic neural language models that involves optimizing a margin-based ranking objective [4]. As that approach is non-probabilistic, it is outside the scope of this paper, though it would be interesting to see whether it can be used to learn competitive word embeddings.

## 4 Evaluating word embeddings

Using word embeddings learned by neural language models outside of the language modelling context is a relatively recent development. An early example of this is the multi-layer neural network of [4] trained to perform several NLP tasks which represented words exclusively in terms of learned word embeddings. [18] provided the first comparison of several word embeddings learned with different methods and showed that incorporating them into established NLP pipelines can boost their performance.

Recently the focus has shifted towards evaluating such representations more directly, instead of measuring their effect on the performance of larger systems. Microsoft Research (MSR) has released two challenge sets: a set of sentences each with a missing word to be filled in [20] and a set of analogy questions [11], designed to evaluate semantic and syntactic content of word representations respectively. Another dataset, consisting of semantic and syntactic analogy questions has been released by Google [10].

In this paper we will concentrate on the two analogy-based challenge sets, which consist of questions of the form "$a$ is to $b$ is as $c$ is to __", denoted as $a : b \to c : ?$ . The task is to identify the held-out fourth word, with only exact word matches deemed correct. Word embeddings learned by neural language models have been shown to perform very well on these datasets when using the following vector-similarity-based protocol for answering the questions. Suppose $\vec{w}$ is the representation vector for word $w$ normalized to unit norm. Then, following [11], we answer $a : b \to c : ?$ , by finding the word $d^*$ with the representation closest to $\vec{b} - \vec{a} + \vec{c}$ according to cosine similarity:

$$d^* = \arg\max_x \frac{(\vec{b} - \vec{a} + \vec{c})^\top \vec{x}}{\|\vec{b} - \vec{a} + \vec{c}\|}. \tag{10}$$

We discovered that reproducing the results reported in [10] and [11] for publicly available word embeddings required excluding $b$ and $c$ from the vocabulary when looking for $d^*$ using Eq. 10, though that was not clear from the papers. To see why this is necessary, we can rewrite Eq. 10 as

$$d^* = \arg\max_x \vec{b}^\top \vec{x} - \vec{a}^\top \vec{x} + \vec{c}^\top \vec{x} \tag{11}$$

and notice that setting $x$ to $b$ or $c$ maximizes the first or third term respectively (since the vectors are normalized), resulting in a high similarity score. This equation suggests the following interpretation of $d^*$: it is simply the word with the representation most similar to $\vec{b}$ and $\vec{c}$ and dissimilar to $\vec{a}$, which makes it quite natural to exclude $b$ and $c$ themselves from consideration.

## 5 Experimental evaluation

### 5.1 Datasets

We evaluated our word embeddings on two analogy-based word similarity tasks released recently by Google and Microsoft Research that we described in Section 4. We could not train on the data used for learning the embeddings in the original papers as it was not readily available. [10] used the proprietary Google News corpus consisting of 6 billion words, while the 320-million-word training set used in [11] is a compilation of several Linguistic Data Consortium corpora, some of which available only to their subscribers.

Instead, we decided to use two freely-available datasets: the April 2013 dump of English Wikipedia and the collection of about 500 Project Gutenberg texts that form the canonical training data for the MSR Sentence Completion Challenge [19]. We preprocessed Wikipedia by stripping out the XML formatting, mapping all words to lowercase, and replacing all digits with 7, leaving us with 1.5 billion words. Keeping all words that occurred at least 10 times resulted in a vocabulary of about 872 thousand words. Such a large vocabulary was used to demonstrate the scalability of our method as well as to ensure that the models will have seen almost all the words they will be tested on. When preprocessing the 47M-word Gutenberg dataset, we kept all words that occurred 5 or more times, resulting in an 80-thousand-word vocabulary. Note that many words used for testing the representations are missing from this dataset, which greatly limits the accuracy achievable when using it. To make our results directly comparable to those in other papers, we report accuracy scores computed using Eq. 10, excluding the second and the third word in the question from consideration, as explained in Section 4.

### 5.2 Details of training

All models were trained on a single core, using minibatches of size 100 and the initial learning rate of $3 \times 10^{-2}$. No regularization was used. Initially we used a validation-set based learning rate adaptation scheme described in [14], which halves the learning rate whenever the validation set

Table 1: Accuracy in percent on word similarity tasks. The models had 100D word embeddings and were trained to predict 5 words on both sides of the current word on the 1.5B-word Wikipedia dataset. Skip-gram(*) is our implementation of the model from [10]. ivLBL is the inverse language model without position-dependent weights. NCE$k$ denotes NCE training using $k$ noise samples.

| | | GOOGLE | | MSR | TIME |
|---|---|---|---|---|---|
| MODEL | SEMANTIC | SYNTACTIC | OVERALL | | (HOURS) |
| SKIP-GRAM(*) | 28.0 | 36.4 | 32.6 | 31.7 | 12.3 |
| IVLBL+NCE1 | 28.4 | 42.1 | 35.9 | 34.9 | 3.1 |
| IVLBL+NCE2 | 30.8 | 44.1 | 38.0 | 36.2 | 4.0 |
| IVLBL+NCE3 | 34.2 | 43.6 | 39.4 | 36.3 | 5.1 |
| IVLBL+NCE5 | 37.2 | 44.7 | 41.3 | 36.7 | 7.3 |
| IVLBL+NCE10 | 38.9 | 45.0 | 42.2 | 36.0 | 12.2 |
| IVLBL+NCE25 | 40.0 | 46.1 | 43.3 | 36.7 | 26.8 |

Table 2: Accuracy in percent on word similarity tasks for large models. The Skip-gram† and CBOW† results are from [10]. ivLBL models predict 5 words before and after the current word. vLBL models predict the current word from the 5 preceding and 5 following words.

| | EMBED. | TRAINING | GOOGLE | | | MSR | TIME |
|---|---|---|---|---|---|---|---|
| MODEL | DIM. | SET SIZE | SEM. | SYN. | OVERALL | | (DAYS) |
| SKIP-GRAM† | 300 | 1.6B | 52.2 | 55.1 | 53.8 | | 2.0 |
| SKIP-GRAM† | 300 | 785M | 56.7 | 52.2 | 55.5 | | 2.5 |
| SKIP-GRAM† | 1000 | 6B | 66.1 | 65.1 | 65.6 | | 2.5×125 |
| IVLBL+NCE25 | 300 | 1.5B | 61.2 | 58.4 | 59.7 | 48.8 | 1.2 |
| IVLBL+NCE25 | 300 | 1.5B | 63.6 | 61.8 | 62.6 | 52.4 | 4.1 |
| IVLBL+NCE25 | 300×2 | 1.5B | 65.2 | 63.0 | 64.0 | 54.2 | 4.1 |
| IVLBL+NCE25 | 100 | 1.5B | 52.6 | 48.5 | 50.3 | 39.2 | 1.2 |
| IVLBL+NCE25 | 100 | 1.5B | 55.9 | 50.1 | 53.2 | 42.3 | 2.9 |
| IVLBL+NCE25 | 100×2 | 1.5B | 59.3 | 54.2 | 56.5 | 44.6 | 2.9 |
| CBOW† | 300 | 1.6B | 16.1 | 52.6 | 36.1 | | 0.6 |
| CBOW† | 1000 | 6B | 57.3 | 68.9 | 63.7 | | 2×140 |
| VLBL+NCE5 | 300 | 1.5B | 40.3 | 55.4 | 48.5 | 48.7 | 0.3 |
| VLBL+NCE5 | 100 | 1.5B | 45.0 | 56.8 | 51.5 | 52.3 | 2.0 |
| VLBL+NCE5 | 300 | 1.5B | 54.2 | 64.8 | 60.0 | 58.1 | 2.0 |
| VLBL+NCE5 | 600 | 1.5B | 57.3 | 66.0 | 62.1 | 59.1 | 2.0 |
| VLBL+NCE5 | 600×2 | 1.5B | 60.5 | 67.1 | 64.1 | 60.8 | 3.0 |

perplexity failed to improve after some time, but found that it led to poor representations despite achieving low perplexity scores, which was likely due to undertraining. The linear learning rate schedule described in [10] produced better results. Unfortunately, using it requires knowing in advance how many passes through the data will be performed, which is not always possible or convenient. Perhaps more seriously, this approach might result in undertraining of representations for rare words because all representation share the same learning rate.

AdaGrad [5] provides an automatic way of dealing with this issue. Though AdaGrad has already been used to train neural language models in a distributed setting [10], we found that it helped to learn better word representations even using a single CPU core. We reduced the potentially prohibitive memory requirements of AdaGrad, which requires storing a running sum of squared gradient values for each parameter, by using the same learning rate for all dimensions of a word embedding. Thus we store only one extra number per embedding vector, which is helpful when training models with hundreds of millions of parameters.

## 5.3 Results

Inspired by the excellent performance of tree-based models of [10], we started by comparing the best-performing model from that paper, the Skip-gram, to its non-hierarchical counterpart, ivLBL without position-dependent weights, proposed in Section 3, trained using NCE. As there is no publicly available Skip-gram implementation, we wrote our own. Our implementation is faithful to the description in the paper, with one exception. To speed up training, instead of predicting all context words around the current word, we predict only one context word, sampled at random using the

Table 3: Results for various models trained for 20 epochs on the 47M-word Gutenberg dataset using NCE5 with AdaGrad. (D) and (I) denote models with and without position-dependent weights respectively. For each task, the left (right) column give the accuracy obtained using the conditional (target) word embeddings. $n$L ($n$R) denotes $n$ words on the left (right) of the current word.

| MODEL | CONTEXT SIZE | GOOGLE | | | | | | MSR | | TIME (HOURS) |
|---|---|---|---|---|---|---|---|---|---|---|
| | | SEMANTIC | | SYNTACTIC | | OVERALL | | | | |
| vLBL(D) | 5L + 5R | 2.4 | 2.6 | 24.7 | 23.8 | 14.6 | 14.2 | 23.4 | 23.1 | 2.6 |
| vLBL(D) | 10L | 1.9 | 2.8 | 22.1 | 14.8 | 12.9 | 9.3 | 20.9 | 9.0 | 2.6 |
| vLBL(D) | 10R | 2.7 | 2.4 | 13.1 | 24.1 | 8.4 | 14.2 | 8.8 | 23.0 | 2.6 |
| vLBL(I) | 5L + 5R | 3.0 | 2.9 | 27.5 | 29.6 | 16.4 | 17.5 | 22.9 | 24.2 | 2.3 |
| vLBL(I) | 10L | 2.5 | 2.8 | 23.5 | 16.1 | 14.0 | 10.1 | 19.8 | 10.1 | 2.3 |
| vLBL(I) | 10R | 2.3 | 2.6 | 16.2 | 24.6 | 9.9 | 14.6 | 10.0 | 20.3 | 2.1 |
| ivLBL(D) | 5L + 5R | 2.8 | 2.3 | 15.1 | 13.0 | 9.5 | 8.1 | 14.5 | 14.0 | 1.2 |
| ivLBL(I) | 5L + 5R | 2.8 | 2.6 | 26.8 | 26.8 | 15.9 | 15.8 | 21.4 | 21.0 | 1.2 |

non-uniform weighting scheme from the paper. Note that our models are also trained using the same context-word sampling approach. To make the comparison fair, we did not use AdaGrad for our models in these experiments, using the linear learning rate schedule as in [10] instead.

Table 1 shows the results on the word similarity tasks for the two models trained on the Wikipedia dataset. We ran NCE training several times with different numbers of noise samples to investigate the effect of this parameter on the representation quality and training time. The models were trained for three epochs, which in our experience provided a reasonable compromise between training time and representation quality.[1] All NCE-trained models outperformed the Skip-gram. Accuracy steadily increased with the number of noise samples used, as did the training time. The best compromise between running time and performance seems to be achieved with 5 or 10 noise samples.

We then experimented with training models using AdaGrad and found that it significantly improved the quality of embeddings obtained when training with 10 or 25 noise samples, increasing the semantic score for the NCE25 model by over 10 percentage points. Encouraged by this, we trained two ivLBL models with position-independent weights and different embedding dimensionalities for several days using this approach. As some of the best results in [10] were obtained with the CBOW model, we also trained its non-hierarchical counterpart from Section 3, vLBL with position-independent weights, using 100/300/600-dimensional embeddings and NCE with 5 noise samples, for shorter training times. Note that due to the unavailability of the Google News dataset used in that paper, we trained on Wikipedia. The scores for ivLBL and vLBL models were obtained using the conditional word and target word representations respectively, while the scores marked with $d \times 2$ were obtained by concatenating the two word representations, after normalizing them.

The results, reported in Table 2, show that our models substantially outperform their hierarchical counterparts when trained using comparable amounts of time and data. For example, the 300D ivLBL model trained for just over a day, achieves accuracy scores 3-9 percentage points better than the 300D Skip-gram trained on the same amount of data for almost twice as long. The same model trained for four days achieves accuracy scores that are only 2-4 percentage points lower than those of the 1000D Skip-gram trained on four times as much data using 75 times as many CPU cycles. By computing word similarity scores using the conditional and the target word representations concatenated together, we can bring the accuracy gap down to 2 percentage points at no additional computational cost. The accuracy achieved by vLBL models as compared to that of CBOW models follows a similar pattern. Once again our models achieve better accuracy scores faster and we can get within 3 percentage points of the result obtained on a cluster using much less data and far less computation.

To determine whether we were crippling our models by using position-independent weight, we evaluated all model architectures described in Section 3 on the Gutenberg corpus. The models were trained for 20 epochs using NCE5 and AdaGrad. We report the accuracy obtained with both conditional and target representation (left and right columns respectively) for each of the models in Ta-

Table 4: Accuracy on the MSR Sentence Completion Challenge dataset.

| MODEL | CONTEXT SIZE | LATENT DIM | PERCENT CORRECT |
|---|---|---|---|
| LSA [19] | SENTENCE | 300 | 49 |
| SKIP-GRAM [10] | 10L+10R | 640 | 48.0 |
| LBL [14] | 10L | 300 | 54.7 |
| IVLBL | 5L+5R | 100 | 51.0 |
| IVLBL | 5L+5R | 300 | 55.2 |
| IVLBL | 5L+5R | 600 | 55.5 |

ble 3. Perhaps surprisingly, the results show that representations learned with position-independent weights, designated with (I), tend to perform better than the ones learned with position-dependent weights. The difference is small for traditional language models (vLBL), but is quite pronounced for the inverse language model (ivLBL). The best-performing representations were learned by the traditional language model with the context surrounding the word and position-independent weights.

**Sentence completion:** We also applied our approach to the MSR Sentence Completion Challenge [19], where the task is to complete each of the 1,040 test sentences by picking the missing word from the list of five candidate words. Using the 47M-word Gutenberg dataset, preprocessed as in [14], as the training set, we trained several ivLBL models with NCE5 to predict 5 words preceding and 5 following the current word. To complete a sentence, we compute the probability of the 10 words around the missing word (using Eq. 4) for each of the candidate words and pick the one producing the highest value. The resulting accuracy scores, given in Table 4 along with those of several baselines, show that ivLBL models perform very well. Even the model with the lowest embedding dimensionality of 100, achieves 51.0% correct, compared to 48.0% correct reported in [10] for the Skip-gram model with 640D embeddings. The 55.5% correct achieved by the model with 600D embeddings is also better than the best single-model score on this dataset in the literature (54.7% in [14]).

## 6 Discussion

We have proposed a new highly scalable approach to learning word embeddings which involves training lightweight log-bilinear language models with noise-contrastive estimation. It is simpler than the tree-based language modelling approach of [10] and produces better-performing embeddings faster. Embeddings learned using a simple single-core implementation of our method achieve accuracy scores comparable to the best reported ones, which were obtained on a large cluster using four times as much data and almost two orders of magnitude as many CPU cycles. The scores we report in this paper are also easy to compare to, because we trained our models only on publicly available data.

Several promising directions remain to be explored. [8] have recently proposed a way of learning multiple representations for each word by clustering the contexts the word occurs in and allocating a different representation for each cluster, prior to training the model. As ivLBL predicts the context from the word, it naturally allows using multiple context representations per current word, resulting in a more principled approach to the problem based on mixture modeling. Sharing representations between the context and the target words is also worth investigating as it might result in better-estimated rare word representations.

**Acknowledgments**

We thank Volodymyr Mnih for his helpful comments.

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

## Footnotes

[1]We checked this by training the Skip-gram model for 10 epochs, which did not result in a substantial increase in accuracy.
