[Reviews · NeurIPS 2013]

Submitted by Assigned_Reviewer_5

The paper proposes the use of noise-contrastive estimation (NCE) to train various neural language models. While the general idea has been proposed before (Mnih and Teh, 2012 in the paper), here NCE is applied to variations of more recent models (Mikolov, 2013a), which had previously relied on tree-structured vocabularies.

The paper shows that the proposed flat (no tree structure) implementations of (Mikolov, 2013a) can achieve similar performance faster.

The models are evaluated on two recent analogy making based similarity tasks (described in Mikolov 2013a). I'm not completely sure how significant the results are as the author's own implementations of (slight variations of) the compared-to models and different training corpora have been used in most of these. Wouldn't it make sense to also include results from the MSR sentence completion challenge?

line 170: "making the training time effectively independent of the vocabulary size". I'm not sure I agree with this statement. Wouldn't that be too good to be true, as it should be harder to get effectively the same number of "negative" samples as the vocabulary size increases? In other words, wouldn't you need more and more noise samples to achieve the same performance, as the vocabulary size increases?

I am wondering if computing word analogies is related in some way the task of computing visual analogies (eg. Susskind, CVPR 2011) as this seems like an analogous problem. So it seems like there should be a non-trivial relationship between the two (like, maybe, the fact that both seem to work efficiently with bilinear models)?

The paper would be a bit easier to follow if it included subsections (for example just before line 130).

Citation style may not conform to nips requirements?

Summary: The paper shows that one can use noise-contrastive estimation to train variations of the language models in (Mikolov 2013a). The novelty might be limited a bit by the fact that NCE has been adapted to language models previously by Mnih and Teh, 2012. The paper presents some encouraging results on analogy making tasks in terms of accuracy and speed.

Submitted by Assigned_Reviewer_8

** Summary of the paper:

Similarly to the work of Minh and Teh ICML2012 the authors use a simple log-bilinear language model trained by noise-contrastive estimation (NCE, Gutmann & Hyvarinen 2012) on large text corpora. The objective here is to learn word embeddings, that are then evaluated on analogy-based word-similarity tasks. Leveraging the fast training afforded by NCE and using a sligthly simpler model than Minh and Teh ICML2012 they are able to outperform state-of-the-art method of Mikolov et al. 2013, using four times less data and an order of magnitude less computing time.


** Additional remarks following the author's rebuttal:

I strongly encourage the authors, in their final version, to emphasize the demonstration of scalability relative to Minh and Teh, as well as their other novel empirical findings in the line of what they did in the first paragraph of their rebuttal.

Regarding the "qualitative comparison" I was thinking of : it is sometimes interesting to visualize 2d or 3d embeddings (eg using t-SNE) of the learned representation for a select sub-part of the vocabulary, as it can sometimes show interesting qualitative differences in the learned structure. While clearly limited and subjective, it is sometimes a good complement to an objective quantitative performance evaluation.

** Evaluation of the paper

This is a well organized, cleraly written and well executed work with convincing experimental results. It is significant in that it demonstrates empirically that noise contrastive estimation allows training simple unnormalized flat models efficiently, that can learn word embeddings as good or better than more sophisticated tree models such as those used by Mikolov et al. Some form of qualitative illustration of differences in learned word embeddings between the approaches (in addition to performance on word-similarity task) would have been welcome.

This paper's weakest aspect is that, except for the difference in application, it is not a very novel approach compared to Minh and Teh ICML2012, that first put forward and experimented with noise contrastive estimation to learn a log-biliniear language model.

There are however a few additional/original, if minor, contributions: a) the simplification of the log-linear model to use vectors instead of matrices for its context weights (associated to a computational gain) b) considering also an "inverse language modeling approach" (predict surrounding context from word)

** Typos

line 303-304: Mikolov et al. should be cited in parentheses here (\citep )

line 369-370: "were obtained the CBOW model" -> "were obtained with the CBOW model"
Summary: ** Summary

+ A clearly written and well executed work that shows how a good word embedding can be learned with a simple langage model and noise contrastive estimation, beating the state-of-the-art compared to more sophisticated tree-based approaches.

- Not much novel compared to Minh and Teh ICML2012 (very minor algorithmic contribution, yet worthy empirical "application" contribution).

Submitted by Assigned_Reviewer_9

Summary:

The paper proposes an efficient approach to learning word embeddings which, when evaluated using three publicly available analogy question challenges, outperform other recently proposed systems (controlled for training time and train set size). The major
contributions appear to be: i) the use of Adagrad (Adaptive Gradient Descent) to accelerate learning; ii) simplification of the pooling layer (from matrix to vector products) in the LBL algorithm of Mnih and Hinton (2007); iii) elimination of hierarchical softmax layer in favor of Noise-contrastive Estimation (NCE) to speed up training. When combined, these result in a reduction in absolute error of 3-9%
relative to the next best reported results for the analogy question challenge sets under consideration when training time and training set size are held constant.


Comments:
The paper is well written but is incremental, has little novelty and is more of an engineering paper. It might be better fit for an NLP venue like EMNLP, NAACL, ACL etc.

Of the three contributions of the paper, I found the last one, most novel, though I wonder how much of the speedup comes from the reduction in time complexity from
the use of NCE and how much from Adagrad (which we know to significantly reduce training times needed to reach a given error rate when applied to acoustic modeling or image classification (see the NIPS and ICML papers that Andrew Ng, Jeff Dean, et al. had in 2012)).

Also, while the performance improvements on the analogy tasks are certainly substantial, I do wonder how useful this is as an evaluation of the quality of the resultant representations; however, as I don't have a better direct evaluation, I can hardly fault the use of this for a testbed and their model certainly improves on the state of the art here.

Additional comments:

- The authors don't compare their embeddings against word embeddings learned using Spectral Learning techniques e.g. CCA, PCA (Dhillon et. al. NIPS 11, ICML 12) as being spectral they are known to be fast. I can see that the authors only want to cater to the Deep Learning audience, but then it significantly decreases the appeal of the paper to the broad NIPS audience.

- The paper looks like a rebuttal to (Mikolov et. al 2013), as that paper gets mentioned ~20 times, including the abstract which narrows the scope and appeal of the paper.

-The authors repeatedly introduce acronyms without explicit definition. NPLM (Neural probabilistic language model) is not defined before first use in the second paragraph of Section 2 (though its meaning can be deduced from the section heading).Similarly, CBOW (Continuous Bag of Words) is not defined before first use in the last paragraph of Section 1. Nor is MSR (Microsoft Research) defined before first use in Table 1. Should somewhere in caption or text indicate that this refers to the Microsoft Research analogy question test set from: Mikolov et al. (2013). "Linguistic regularities in continuous space word representations"

-Precisely how the embeddings are learned is a bit obscure. In Section 3, where they introduce LBL models, it seems to assume that we have an embedding prior to training the LBL model for prediction. Going back to the Mnih and Hinton papers cited in the references helps a little, but it still is not entirely clear. I assume they jointly learn the embedding matrix with the LBL model?

-It would be nice to compare LBL models trained with Adagrad but not NCE to models trained with both to see to what extent the performance improvements come from each.
Summary: Mostly, an engineering paper, which improves upon word embeddings recently proposed method by (Mikolov et. al.). Lacks in mathematical novelty and empirical improvements have too narrow an appeal.

**I have read the author rebuttal and my recommendation remains the same.**
Author Feedback

Author rebuttal: We thank the reviewers for their comments and suggestions.

As requested by the reviewers, we evaluated our models on the MSR Sentence Completion dataset. The accuracy scores for the ivLBL model trained on the Gutenberg dataset using 100/300/600D embeddings are 51.0/55.2/55.5% correct respectively. These compare quite favourably to 48.0% correct reported for the Skip-gram model with 640D features by Mikolov et al. (2013a). The 55.5% score achieved by the 600D model is also better than the best single model score on this dataset in the literature (54.8% in Mnih and Teh (2012)).

We believe that the main contribution of the paper is the demonstration that NCE training of lightweight language models is a highly efficient way of learning high-quality word representations. Consequently, results that used to require very considerable hardware and software infrastructure can now be obtained on an a single desktop with minimal programming effort and using less time and data. In contrast to Mnih and Teh (2012), who showed that NCE can be used to train language models that make accurate predictions, we demonstrated that NCE training also produces effective word representations. Our paper also shows that for representation learning, using 5 noise samples in NCE can be sufficient for obtaining competitive results, which is not the case in language modelling. And finally, we applied NCE on a far larger scale than was done before, using 30 times more training data and a 10 times larger vocabulary.


Reviewer_5:

We agree that it would have been preferable to use the Skip-gram implementation and the dataset from Mikolov et al. (2013a). However, as both of them were not publicly available, we had to write our own implementation and use freely-available datasets. Note that our choice of datasets makes it easy to compare to our results directly.

Though it is indeed likely that the number of noise samples needs to grow with the vocabulary size to maintain the same level of performance, empirically this growth seems slow enough to be negligible. Mnih and Teh (2012) used 25 noise samples to train models with 10K- and 80K-word vocabularies, and we use the same number of samples with a 872K-word vocabulary. Moreover, results in Tables 1 and 2 show that very good representations can be learned using just 5 noise samples.

While there are considerable similarities between the word analogies considered in our paper and the visual analogies from Susskind et al. (2011), the tasks are approached in very different ways. Susskind et al. (2011) model the relationship between images explicitly by training on pairs of related ("analogous") images, while language models such as ours learn from unstructured text, with word analogies only used to evaluate the resulting representations.

We will break up the longer sections into subsections for improved readability. As for the citation style, the official NIPS style file (nips2013.tex) seems to allow any citation style, as long as it is used consistently.


Reviewer_8:

We are not sure what kind of qualitative illustration the reviewer is referring to. We believe that the word analogy tasks (along with the MSR sentence completion task) are a reasonable task-independent evaluation, but we would welcome suggestions of other evaluation techniques.


Reviewer_9:

As the results in Table 1 were obtained without using AdaGrad, they demonstrate that NCE training alone is sufficient to obtain substantial speedups over tree-based training. In our experience, compared to NCE, AdaGrad provides only a modest speedup for training LBL-like models and is more useful as an effective learning rate adaptation method. We also tried using AdaGrad for the tree-based Skip-gram model and found that it did not perform better than the simple linear learning rate annealing schedule used in the paper.

The spectral word embedding estimation techniques are very unlikely to scale to the dataset and representation sizes considered in the paper. The spectral method of Dhillon et al. (2012) took 3 days to learn 50-dimensional embeddings on a 63-million word dataset with a 300k-word vocabulary, which suggests that it is far slower our method. Furthermore, the running time of their CCA-based method scales at least quadratically in the embedding dimensionality (in contrast to the linear scaling of our method), making it ill-suited for learning high-dimensional representations.

Word embeddings in neural language models are always learned jointly with other model parameters, which is suggested by their inclusion in the parameter vector theta on line 76. We will clarify this point in the paper.

We will make sure all acronyms are clearly defined before use.